# Effects of Verapamil and Diltiazem on the Pharmacokinetics and Pharmacodynamics of Rivaroxaban

**DOI:** 10.3390/pharmaceutics11030133

**Published:** 2019-03-19

**Authors:** Minsoo Kim, Heebin Son, Keumhan Noh, Eunyoung Kim, Beom Soo Shin, Wonku Kang

**Affiliations:** 1College of Pharmacy, Chung-Ang University, Seoul 06974, Korea; km4355@naver.com (M.K.); amybin2@naver.com (H.S.); eykimjcb777@cau.ac.kr (E.K.); 2Department of Pharmaceutical Sciences, Leslie Dan Faculty of Pharmacy, University of Toronto, 144 College Street, Toronto, ON M5S 3M2, Canada; keumhan.noh@utoronto.ca; 3School of Pharmacy, Sungkyunkwan University, Suwon 16419, Korea

**Keywords:** drug–drug interaction, PK/PD modeling, prothrombin time, rivaroxaban, verapamil

## Abstract

Concomitant use of rivaroxaban with non-dihydropyridine calcium channel blockers (non-DHPs) might lead to an increase of systemic rivaroxaban exposure and anticoagulant effects in relation to the inhibition of metabolic enzymes and/or transporters by non-DHPs. This study was designed to evaluate the effects of verapamil and diltiazem on the pharmacokinetics and the prolongation of prothrombin time of rivaroxaban in rats. The data were analyzed using a pharmacokinetic/pharmacodynamics (PK/PD) modeling approach to quantify the influence of verapamil. Verapamil increased the systemic exposure of rivaroxaban by 2.8-fold (*p* <0.001) which was probably due to the inhibition of efflux transportation rather than metabolism. Prothrombin time was also prolonged in a proportional manner; diltiazem did not show any significant effects, however. A transit PK model in the absorption process comprehensively describes the double-peaks of rivaroxaban plasma concentrations and the corresponding change of prothrombin time with a simple linear relationship. The slope of prothrombin time vs. rivaroxaban plasma concentration in rats was retrospectively found to be insensitive by about 5.4-fold compared to than in humans. More than a 67% dose reduction in rivaroxaban is suggested in terms of both a pharmacokinetic point of view, and the sensitivity differences on the prolongation of prothrombin time when used concomitantly with verapamil.

## 1. Introduction

Rivaroxaban is a specific, competitive factor Xa inhibitor, directly acting on factor Xa. Not only does it inhibit the activity of factor Xa, but it also affects prothrombinolytic enzymes and free Xa bound to fibrin, thereby inhibiting thrombin generation [1]. In 2008, the drug was approved for the prevention of venous thromboembolism in adults undergoing major orthopedic surgery. Currently, it is used to reduce the risk of stroke and systemic embolism in nonvalvular atrial fibrillation patients and to prevent and treat deep vein thrombosis and pulmonary embolisms. When used in combination with aspirin or clopidogrel, the incidence of atherosclerotic events is also known to be reduced [2,3]. 

Because rivaroxaban is a substrate of cytochrome P450 3A4 (CYP3A4) and P-glycoprotein (P-gp) [4], drug–drug interactions with concomitant drugs that may influence the functions of CYP3A4 and P-gp should be considered. For example, ketoconazole (400 mg, qd) significantly increased the systemic exposure of rivaroxaban [5], whereas rifampicin, a CYP3A4 inducer, resulted in a 50% reduction in the systemic exposure of rivaroxaban [2]. Non-dihydropyridine calcium channel blockers (non-DHPs), verapamil and diltiazem, are also well known to be CYP3A4/P-gp inhibitors [6,7] and have been frequently prescribed in patients with arterial fibrillation [4,5]. Therefore, concomitant use of rivaroxaban with these non-DHPs might lead to an increase in systemic rivaroxaban exposure and anticoagulant effects. However, when compared with rivaroxaban use alone, the adjusted incidence rate for major bleeding did not significantly increase when patients with non-valvular arterial fibrillation were treated with concurrent use of both drugs [8]. Additionally, concurrent use of both drugs in arterial fibrillation patients was not associated with an increased risk of stroke/non-CNS systemic embolism (*p* < 0.11), or non-major clinically relevant bleeding (*p* < 0.087), however, an increased risk of major bleeding and intracranial hemorrhage was strongly related [5]. Additionally, a recent report outlined that in patients with mild renal insufficiency, verapamil increased the systemic exposure of rivaroxaban and bleeding risk so that modification of the recommended dosage was suggested [9].

To date, the drug–drug interactions between rivaroxaban and verapamil/diltiazem are still unclear, especially from a quantitative point of view. Therefore, this study was designed to evaluate the effects of verapamil and diltiazem on the pharmacokinetics and pharmacodynamics of rivaroxaban in rats. Rivaroxaban plasma concentrations and prothrombin time were serially measured following oral administration of rivaroxaban in the presence, or absence, of either verapamil or diltiazem. Data were analyzed by using a pharmacokinetic/pharmacodynamic (PK/PD) modeling approach to quantify the influence of concurrent drugs.

## 2. Materials and Methods

### 2.1. Materials

Rivaroxaban (purity > 99%) was purchased from ChemScene (Monmouth Junction, NJ, USA). Verapamil and diltiazem were kindly provided by Ilsung Pharmaceuticals (Seoul, South Korea) and CJ Healthcare (Seoul, South Korea), respectively. Flufenamic acid, used as an internal standard, was purchased from Sigma-Aldrich (St. Louis, MN, USA). All other chemicals and solvents were of the highest analytical grade available.

### 2.2. Animal Study

The animals used in this study were 15 Sprague Dawley rats (8-week-old males, 280–300 g). The animal room was maintained at a temperature of 23 ± 3 °C, relative humidity of 50 ± 10% with 10–20 air changes/h, and light intensity of 300 Lux with a 12 h light/dark cycle. The study protocol was approved by the Institutional Animal Care and Use Committees (IACUC) at Chung-Ang University. All animals used in this study were cared for in accordance with the principles outlined in the National Institutes of Health Guide for the Care and Use of Laboratory Animals.

Rats were randomized into three groups: a control group, a verapamil-treated group, and a diltiazem-treated group, each consisting of five animals. Rivaroxaban (2 mg/kg) was administrated in the presence, or absence, of either verapamil (25 mg/kg) or diltiazem (30 mg/kg). All drugs were suspended in corn oil and administered orally.

To measure rivaroxaban plasma concentrations, heparinized blood samples (100 µL) were taken at 0.5, 1, 1.5, 2, 3, 4, 5, 6, 8, 10, 12, and 15 h after administration via the subclavian vein. After centrifugation (17,000 rpm, 10 min), plasma samples were stored at –70 °C until analysis, using 30 μL of plasma. For measuring prothrombin time, blood samples (450 µL) were taken at 0, 1.5, 3, 6 and 10 h, in tubes containing 3.2% sodium citrate. After centrifugation (3000 rpm, 10 min), plasma samples were stored at 4 °C until analysis and prothrombin time was measured within 24 h.

### 2.3. Rivaroxaban Plasma Concentrations Measured by LC-MS/MS

Rivaroxaban plasma concentrations were determined using high-performance liquid chromatography coupled to a tandem mass spectrometry (LC-MS/MS). Acetonitrile (90 μL) including flufenamic acid (10 ng/mL) was added to the plasma samples (30 μL). The mixture was vigorously vortexed for 10 s and centrifuged at 17,000 rpm for 10 min. A 5-μL sample of the supernatant was injected onto the LC-MS/MS.

Rivaroxaban and the internal standard were quantified using an API 4000 LC-MS/MS system (ABSCIEX, Framingham, MA, USA) equipped with an electrospray ionization interface. The compounds were purified on a reversed-phase column (Atlantis^®^ T3, 50 × 2.1 mm internal diameter, 3 µm particle size; Waters) at 30 °C. The mobile phase consisted of distilled water and acetonitrile (4:6, v/v) including 0.1% formic acid and ran at a flowrate of 0.2 mL/min using an HP 1260 series pump (Agilent, Wilmington, DE, USA) over a period of 5 min.

### 2.4. Measurement of Prothrombin Time

Prothrombin time was measured using a blood coagulation analyzer (ACL 7000, Instrumentation Laboratory, Bedford, MA, USA) at Korea Conformity Laboratories (Incheon, Korea) using PT recombiPlasTin 2G reagent (Instrumentation Laboratory, Bedford, MA, USA). 

### 2.5. Model Independent Data Analysis

Time courses of rivaroxaban plasma concentrations were used to calculate the pharmacokinetic parameters: peak concentration (*C_max_*) and time to *C_max_* (*T_max_*) were directly obtained from the individual time courses; elimination rate constant (k) was obtained by linear regression from the log-transformed rivaroxaban plasma concentrations at the terminal phase; area under the plasma concentration-time curve (*AUC_t_*) was calculated by the trapezoidal rule, and the concentration at the last sampling time (*C_last_*)/*k* was added to obtain AUC to infinite (*AUC_inf_*); and clearance (*CL*) was calculated by *dose*/*AUC_inf_*. 

The time courses of prothrombin time were used to determine pharmacodynamic parameters: peak effect observed (*E_max,__observed_*) was directly obtained from the individual time courses; and the area under the effect-time curve (*AUEC_t_*) was calculated by the trapezoidal rule. 

### 2.6. Model Dependent Data Analysis

A sequential PK/PD modeling approach was performed to build the best model for rivaroxaban concentration and effect-data over time. A PK model to fit the time courses of rivaroxaban plasma concentrations was constructed, and a linear PD model was linked to describe the change of prothrombin time after fixing the PK parameters [10]. 

The PK model development was an iterative process with regard to both the underlying data sets and the selected model structures. Because the time courses of rivaroxaban plasma concentrations represented a double-peak phenomenon, a bypass route was needed to fit the second peak, as shown in Figure 1, i.e., the delayed absorption (*k_delay_*) was incorporatedf following a transit of 3–8 compartments (N_i_, Δ), and the total absorption was devided into (1-*f*)*k_a_* and *f·k_delay_* at depot compartment (1). A 1-compartment and the first order elimination (*k_e_*) were good enough to describe the distribution and elimination processes of rivaroxaban. 

A proportional model without an intercept was used to represent the relationship between rivaroxaban plasma concentrations and prothrombin time. No delay parameter was needed because the time profiles of prothrombin time were nearly superimposed with corresponding changes of plasma concentrations (Figure 2).

The differential equations that describe changes in the amounts of rivaroxaban in the compartments were given by Equations (1)–(4).
(1)$$\frac{dx_{1}}{\mathrm{dt}}=\;-\left(1-f\right)\cdot k_{a}\cdot x_{1}-f\cdot k_{delay}\cdot x_{1}$$
(2)$$\frac{dx_{2}}{\mathrm{dt}}=\;\left(1-f\right)\cdot k_{a}\cdot x_{1}+k_{delay}\cdot x_{i}-k_{e}\cdot x_{2}$$
(3)$$\frac{dx_{3}}{\mathrm{dt}}=\;f\cdot k_{delay}\cdot x_{1}-k_{delay}\cdot x_{3}$$
(4)$$\frac{dx_{i}}{\mathrm{dt}}=\;f\cdot k_{delay}\cdot x_{i-1}-k_{delay}\cdot {\;x}_{i}$$

The plasma concentrations induce the effect *E*(*t*) by a linear model (Equation (5)).
(5)$$E\left(t\right)=slope\cdot \frac{x_{2}}{V_{d}}$$

Equations (1)–(4) were solved numerically and fitted to the data by means of maximum likelihood estimation using the ADAPT 5 software package (Biomedical Simulations Resource, Los Angeles, CA, USA) with the error model as follows:(6)$$\hat{C}\left(t_{i}\right)=\;C\left(t_{i}\right)+\varepsilon_{i}$$
(7)$$\mathrm{var}\left[\varepsilon_{i}\left(t\right)\right]=(\sigma_{0}+\sigma_{1}C{\left(t_{i})\right)}^{2}$$
where (*t*) denotes the measured concentration and Ĉ(*t*) = x_2_(*t*)/*V* is the model prediction. The pharmacokinetic parameters were fixed in fitting the effect-data to estimate slope, using the same error model.

To evaluate the model’s goodness-of-fit, visual examination was first performed to assess how well the measured concentrations and the predicted model values visually overlapped. Several fitting models were statistically evaluated using the likelihood ratio test, and comparisons were made between each one [11]. Statistical significance was assessed at a level of *p* < 0.05. The simplest model that adequately explains the given data, according to the principle of parsimony, was selected. The following information provided by ADAPT 5 was used to evaluate the goodness-of-fit and the quality of parameter estimates: Coefficients of variation (CV) of parameter estimates, parameter correlation matrix, sum of squares of residuals, and visual examination of the distribution of residuals.

In the first step, individual data sets were independently fitted and each pharmacokinetic and pharmacodynamic parameter was represented as the mean and variance.

### 2.7. Statistical Analysis

The estimated parameters obtained from the control and non-DHP treated groups were statistically compared using Student’s *t*-test. Statistical significance was assessed at a level of *p* < 0.05.

## 3. Results and Discussion

### 3.1. Pharmacokinetics of Rivaroxaban in the Presence of Verapamil or Diltiazem

Mean time courses of rivaroxaban plasma concentrations in the absence and presence of verapamil or diltiazem are represented in Figure 3, and pharmacokinetic parameters of rivaroxaban are listed in Table 1.

Rivaroxaban concentrations peaked in plasma 2.8 h after oral administration. The mean *C_max_* and *AUC_inf_* were 373 ng/mL and 2699 ng·h/mL, respectively, and *CL/F* is 774 mL/h/kg. Verapamil increased *C_max_* and *AUC_inf_* of rivaroxaban by 2.9- (*p* < 0.01) and 2.8-fold (*p* < 0.001), respectively, and clearance decreased by 63%, compared with the control group. In contrast, although diltiazem significantly increased the *C_max_* of rivaroxaban by 1.4-fold (*p* < 0.05), no statistical differences were found in clearance and *AUC_inf_*. As expected, verapamil, a strong inhibitor of CYP/P-gp, exerted much greater effect on the systemic exposure of rivaroxaban than diltiazem, a moderate inhibitor. The increase of rivaroxaban exposure by verapamil may be attributed to the changes of absorption and/or disposition related to P-gp, rather than metabolism, because there was no significant difference in the elimination rate of rivaroxaban (see model parameters below). The present results are comparable to a previous publication by Frost et al.: diltiazem represents a weaker and non-significant action on the systemic exposure of apixaban as compared with ketoconazole, a strong inhibitor of CYP3A4/P-gp, and both drugs did not affect the terminal half-life of apixaban [7].

Although a multiple-peaks phenomenon was observed in the mean time courses of rivaroxaban plasma concentrations (Figure 3), double-peaks are more profound in individual data sets, and the multiple peaks are a result of inter-individual variability on the times of those peaks. The double-peak phenomenon following oral administration can be explained by three major causes: Entero-hepatic circulation, delayed gastric emptying, and/or absorption at various sites in the gastrointestinal tracts.

### 3.2. Prothrombin Time Following Oral Administration of Rivaroxaban in the Presence of Verapamil or Diltiazem

Time profiles of prothrombin time and percentage increase following oral administration of rivaroxaban in the presence, or absence, of either verapamil or diltiazem are shown in Figure 4A,B, respectively, and pharmacodynamic parameters are listed in Table 2.

The mean maximum percentage increase (*E_max_*) of prothrombin time was 30% when rivaroxaban was given alone, and the mean area under the effect curve (*AUEC*) was 188%∙h. While diltiazem did not represent any significant changes in either parameter, verapamil significantly increased *E_max_* and *AUEC* by 2.5- and 2.6-fold (*p* < 0.01), respectively. 

The prolonged prothrombin time by co-administration of verapamil was probably due to the increment of the systemic exposure of rivaroxaban. As shown in Figure 5, prothrombin time proportionally increased with increasing rivaroxaban plasma concentrations, with the slopes of both curves clearly superimposed.

### 3.3. PK/PD Modeling

The present PK/PD model developed describes both rivaroxaban plasma concentrations and percentage increase of prothrombin time over time very well.

The PK/PD parameter estimates for rivaroxaban in the presence or absence of verapamil are given in Table 3. 

Initially, for rivaroxaban alone, 40% of the dose was estimated to be absorbed, contributing to the first peak, with the remainder appearing in circulation with 6.4 h of transit time. On one hand, verapamil significantly accelerated the absorption rate by 62% (*p* < 0.01) and shortened the transit time by 28% (*p* < 0.05), which, in part, may be attributed to the increase of systemic exposure of rivaroxaban. On the other hand, the 2.6-fold decrease (*p* < 0.01) of volume of distribution seems to be inversely reflected by the 2.8-fold increase of *AUC*, as shown in the model independent analysis above. Although verapamil is well known as a strong P-gp inhibitor, the current model could not fully characterize the influence of the drug, which is a limitation of a compartment modeling approach due to the oversimplification of complex systems in vivo. Nevertheless, attention should be paid to the significant action of verapamil on the absorption of rivaroxaban via the inhibition of the transporter. A parameter that was not affected by verapamil was the elimination rate constant (0.85 h^−1^), suggesting that the increment of the extent absorbed may not be due to the metabolic inhibition of rivaroxaban by verapamil.

There was no difference in the changes of percentage increase of prothrombin time corresponding to the rivaroxaban concentration changes (slope) in the presence of verapamil, which means that verapamil may not affect the sensitivity of the phamamacodynamic action of rivaroxaban, a phenomenon reflected by the results illustrated in Figure 5. Although the increase of prothrombin time was remarkably magnified by verapamil, it likely stemmed from the increment of plasma concentrations. Therefore, the linear relationship between plasma concentration and effect still remains at the present combination and dose. Because changes of prothrombin time nearly overlapped with the change of corresponding plasma concentrations (Figure 2), no effect compartment for explaining any delayed action was needed. Additionally, this relationship should be observed because the target of rivaroxaban, Factor Xa, exists in the central compartment.

Kubitza et al. recently reported a linear relationship between rivaroxaban plasma concentration and relative prothrombin time in a healthy pediatric population, with no differences found in results obtained from adults [12]. In the present study, prothrombin times were relatively rescaled to allow direct comparisons with the data obtained by Kubitza et al., namely, the percentage increases of prothrombin time were switched to the ratio of the baseline level: the slope of relative prothrombin time vs. rivaroxaban plasma concentration was approximately 0.0029 in humans, which is 5.4-fold more sensitive than in rats (0.00054).

The quality of the present PK/PD model was evaluated by a visual predictive check, as shown in Figure 6. A 1000-iteration cycle was conducted by Monte Carlo simulation, and a 90% confidence interval was obtained. Few data points appeared outside the confidence interval, indicating that the developed model can characterize both PK and PD data sets simultaneously very well and allow excellent, reliable predictions.

To date, data analysis of rivaroxaban by using PK/PD modeling approaches have mainly been conducted in relation to ROCKET AF trials [13,14]. A few studies have addressed the quantitative pharmacology of the drug in patients with acute coronary syndromes [15] and atrial fibrillation being treated for stroke prevention [16], and those undergoing major orthopedic surgery [17]. All previous studies used population PK/PD modeling with simple one-compartment models due to the characteristics in clinical situations, such as limited blood sampling. The present model comprehensively describes the double-peak phenomenon of time courses of plasma concentrations of the drug and the relationship with prothrombin time in rats, while also measuring the action of verapamil co-administration in quantitative aspects for the first time. While a one-third dose reduction is suggested in terms of a pharmacokinetic point of view, when both drugs are used concomitantly, they could be elevated further, considering the inter-species differences of sensitivity on the prolongation of prothrombin time.

## 4. Conclusions

A transit model in the absorption process comprehensively describes the double-peaks of rivaroxaban plasma concentrations and the corresponding changes in pharmacodynamic effect on prothrombin time in rats. The effects of verapamil on the PK and PD parameters of rivaroxaban were quantified, with the results indicating that a dose reduction is required when used together. The sensitivity of the pharmacological action of rivaroxaban was compared between humans and rats. The clinical significance of the present results should be taken into further account, including the drug–drug interactions between rivaroxaban and P-gp inhibitors, in terms of drug transportations.

## Figures and Tables

**Figure 1 pharmaceutics-11-00133-f001:**
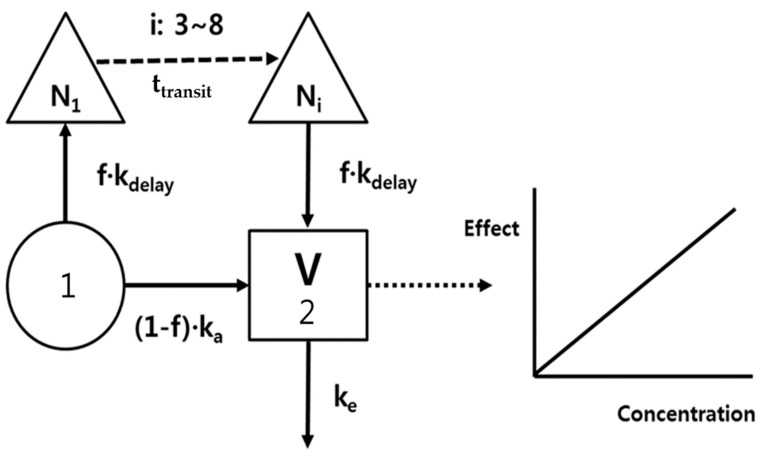
Rivaroxaban pharmacokinetic/pharmacodynamic (PK/PD) model; *k_a_*, absorption rate constant; *k_delay_*, absorption rate constant from transit compartment; *f*, absorption fraction; *N_i_*, number of transit compartments (i); *V*, volume of distribution; and *k_e_*, elimination rate constant.

**Figure 2 pharmaceutics-11-00133-f002:**
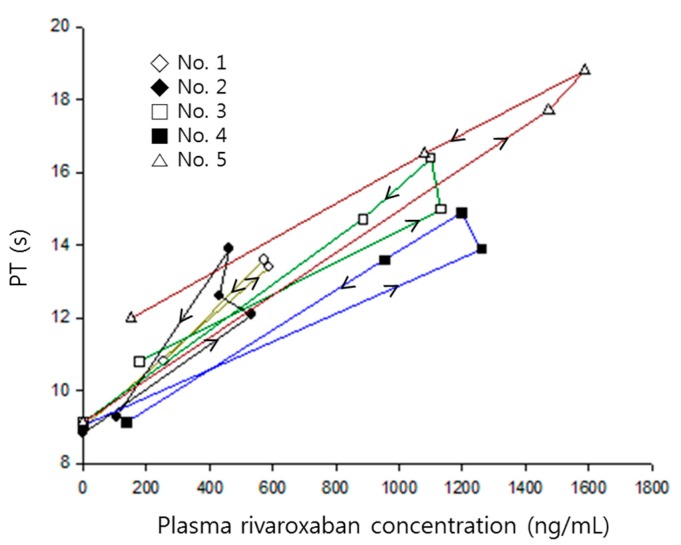
Prothrombin time versus rivaroxaban plasma concentrations in each rat numbered 1–5, when receiving rivaroxaban (2 mg/kg) together with verapamil (25 mg/kg). Arrows indicate the direction of time courses.

**Figure 3 pharmaceutics-11-00133-f003:**
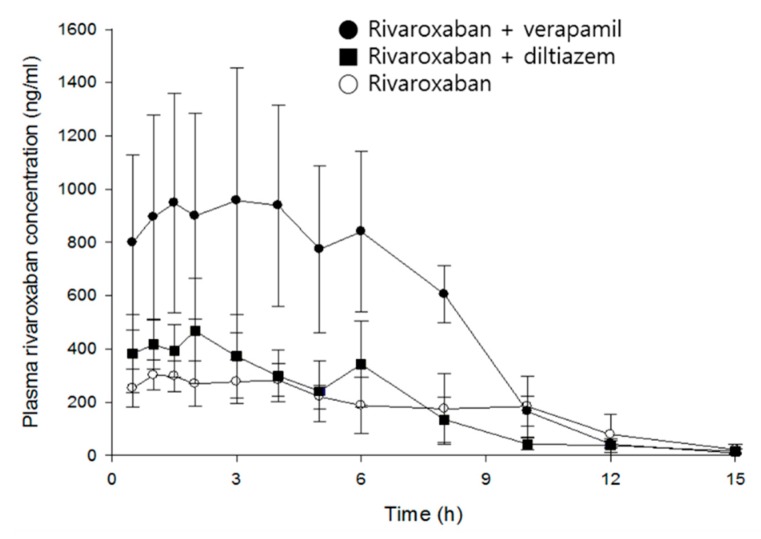
Mean plasma concentration–time profiles of rivaroxaban after an oral administration of rivaroxaban (2 mg/kg) in the presence, or absence, of either verapamil (25 mg/kg) or diltiazem (30 mg/kg) in rats (mean ± SD, *n* = 5).

**Figure 4 pharmaceutics-11-00133-f004:**
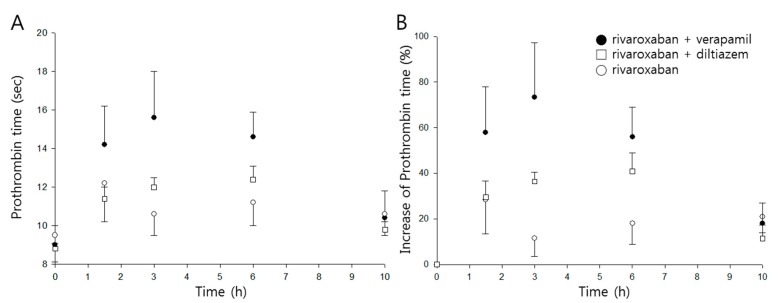
Time courses of prothrombin time (**A**) and percentage increase of prothrombin time (**B**) following oral administration of rivaroxaban (2 mg/kg, ○) in the presence, or absence, of verapamil (25 mg/kg, ●) or diltiazem (30 mg/kg, □) in rats (mean ± SD, *n* = 5).

**Figure 5 pharmaceutics-11-00133-f005:**
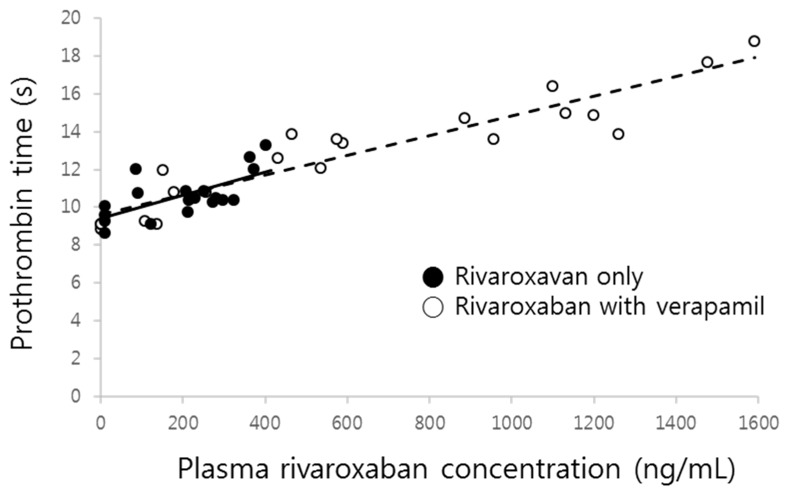
Prothrombin time vs. rivaroxaban plasma concentration in the absence (●, solid line) and presence (○, dashed line) of verapamil.

**Figure 6 pharmaceutics-11-00133-f006:**
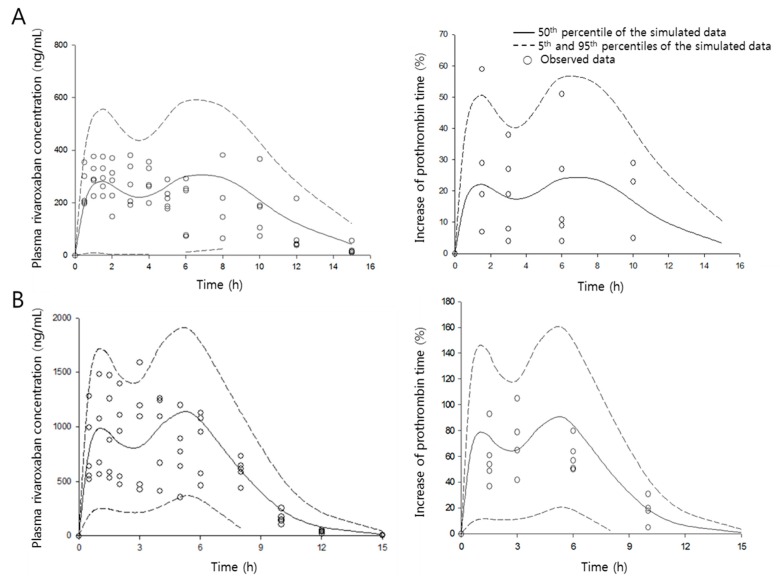
Visual predictive check of the pharmacokinetic/pharmacodynamic model. The observed plasma concentration of rivaroxaban in the absence (**A**) and presence (**B**) of verapamil. Circles are the observed data, and the solid and dashed lines indicate 50th, and 5th and 95th percentiles of confidence intervals, respectively.

**Table 1 pharmaceutics-11-00133-t001:** Pharmacokinetic parameters of rivaroxaban following an oral administration of rivaroxaban (2 mg/kg) in the presence, or absence, of either verapamil (25 mg/kg) or diltiazem (30 mg/kg) (mean (relative standard deviation, %), n = 5). * *p* < 0.05, ** *p* < 0.01, *** *p* < 0.001 compared to rivaroxaban only.

Parameter	Rivaroxaban	Rivaroxaban + Verapamil	Rivaroxaban + Diltiazem
***C_max_*** **(ng/mL)**	373 (3)	1079 (39) **	532 (31) *
***T_max_* (h)**	2.8 (111)	4.0 (63)	1.6 (44)
***AUC_inf_* (ng/mL)**	2699 (25)	7606 (30) ***	2915 (27)
***CL/F* (mL/h/kg)**	774 (22)	286 (35) ***	723 (24)

**Table 2 pharmaceutics-11-00133-t002:** Maximum percentage increase and area under the percentage increase vs. time curve of prothrombin time following oral administration of rivaroxaban (2 mg/kg) in the presence, or absence, of either verapamil (25 mg/kg) or diltiazem (30 mg/kg) (mean (relative standard deviation, %), n = 5). * *p* < 0.01 compared to rivaroxaban only.

Parameter	Rivaroxaban	Rivaroxaban + Verapamil	Rivaroxaban + Diltiazem
***E_max_*** **(%)**	30 (11)	75 (14) *	40 (7)
***AUEC*** **(%** **∙h)**	188 (9)	491 (10) *	247 (14)

**Table 3 pharmaceutics-11-00133-t003:** PK/PD model parameters of rivaroxaban in the presence or absence of verapamil (mean (relative standard deviation, %), *n* = 5). * *p* < 0.05, ** *p* < 0.01.

Parameter	Rivaroxaban	Rivaroxaban + Verapamil
***k_a_* (h^−1^)**	0.63 (22)	1.02 (19) **
***V/F* (mL/kg)**	960 (37)	369 (37) **
***k_e_* (h^−1^)**	0.85 (27)	0.80 (19)
***f* (fraction)**	0.61 (30)	0.61 (14)
***t_transit_* (h)**	6.37 (21)	4.55 (15) *
***Slope* (%·mL/ng)**	0.08 (44)	0.08 (24)

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
