# Peer review of "Effects of Verapamil and Diltiazem on the Pharmacokinetics and Pharmacodynamics of Rivaroxaban"

_pharmaceutics, 2019, doi:10.3390/pharmaceutics11030133_

Round 1

Reviewer 1 Report

 Effects of verapamil and diltiazem on PK/PD of Rivaroxaban were conducted in rat model. In general the variation of results is still high due to the limited number of rats for each group. also the authors showed differences in control and verapamil group, but not diltiazem, one reason could be small sample size, but if is the truth, the authors should provide pieces of evidence to answer this question, the invitro studies may be additional, how metabolism enzymes or transporters may play the role in this effects, especially the differences between verapamil and diltiazem.

Author Response

Effects of verapamil and diltiazem on PK/PD of Rivaroxaban were conducted in rat model. In general the variation of results is still high due to the limited number of rats for each group. also the authors showed differences in control and verapamil group, but not diltiazem, one reason could be small sample size, but if is the truth, the authors should provide pieces of evidence to answer this question, the in vitro studies may be additional, how metabolism enzymes or transporters may play the role in this effects, especially the differences between verapamil and diltiazem.

→ While the variation is not so high in control group and treatment group with diltiazem, it seems to be relatively high in the treatment group with verapamil, which is because double peaks appears more profoundly in the individual data set (Figure 6), when verapamil co-administered, and the times of two peaks are different from rat to rat as mentioned in Section 3.1.

 The differences between verapamil and diltiazem on the pharmacokinetics of rivaroxaban are described at the end of the second paragraph in Section 3.1 as follows:

“As expected, verapamil, a strong inhibitor of CYP/P-gp, exerts much greater effect on the systemic exposure of rivaroxaban than diltiazem, a moderate one. The increase of rivaroxaban exposure by verapamil may be attributed to the changes of absorption and/or disposition related to P-gp rather than metabolism, because there was no significant difference in the elimination rate of rivaroxaban (see model parameters below). The present results are comparable to a previous publication by Frost et al.: diltiazem represents a weaker and no significant action on the systemic exposure of apixaban than ketoconazole, a strong inhibitor of CYP3A4/P-gp, and both drugs did not affect the terminal half-life of apixaban [7].”

7. C.E., Frost, W., Byon, Y., Song, J., Wang, A.E., Schuster, R.A., Boyd, D., Zhang, Z., Yu, C., Dias, A., Shenker, F., LaCreta. Effect of ketoconazole and diltiazem on the pharmacokinetics of apixaban, an oral direct factor Xa inhibitor. Br J Clin Pharmacol 2015, 79, 838-846.

Reviewer 2 Report

Dear Authors,

I read your work with much interest and find it a relevant research article. However, i have 3 comments towards improving the overall quality and conclusions from this research work:

1) Please explain the the lack of Diltiazem effect on Rivaroxaban, in terms of AUC and Cmax. Contrast the differences between Verapamil and Diltiazem that may be responsible for this difference in effect on Rivaroxaban.

2) Does different doses of Rivaroxaban when combined with Verapamil produce comparable effects on the pharmacokinetics of Rivaroxaban?

3) Please mention the number of times the animal experiments were repeated (with comparable results)

Good-luck!

Author Response

1) Please explain the the lack of Diltiazem effect on Rivaroxaban, in terms of AUC and Cmax. Contrast the differences between Verapamil and Diltiazem that may be responsible for this difference in effect on Rivaroxaban.

The differences between verapamil and diltiazem on the pharmacokinetics of rivaroxaban are described at the end of the second paragraph in Section 3.1 as follows:

“As expected, verapamil, a strong inhibitor of CYP/P-gp, exerts much greater effect on the systemic exposure of rivaroxaban than diltiazem, a moderate one. The increase of rivaroxaban exposure by verapamil may be attributed to the changes of absorption and/or disposition related to P-gp rather than metabolism, because there was no significant difference in the elimination rate of rivaroxaban (see model parameters below). The present results are comparable to a previous publication by Frost et al.: diltiazem represents a weaker and no significant action on the systemic exposure of apixaban than ketoconazole, a strong inhibitor of CYP3A4/P-gp, and both drugs did not affect the terminal half-life of apixaban [7].”

7. C.E., Frost, W., Byon, Y., Song, J., Wang, A.E., Schuster, R.A., Boyd, D., Zhang, Z., Yu, C., Dias, A., Shenker, F., LaCreta. Effect of ketoconazole and diltiazem on the pharmacokinetics of apixaban, an oral direct factor Xa inhibitor. Br J Clin Pharmacol 2015, 79, 838-846.

2) Does different doses of Rivaroxaban when combined with Verapamil produce comparable effects on the pharmacokinetics of Rivaroxaban?

We believe ‘Yes’, because verapamil exerts similar consequences to different doses of rivaroxaban as long as the dose is within a range representing a linear kinetics.

3) Please mention the number of times the animal experiments were repeated (with comparable results)

→ This comment seems to be regarding the number of animals in each group. Five animals were used for each group.  

Reviewer 3 Report

Summary This study examined the effects of the non-dihydropyridine calcium channel blockers verapamil and diltiazem on the pharmacokinetics and pharmacodynamics of rivaroxaban in rats. Verapamil increased the systemic exposure of rivaroxaban by 2.8-fold (p<0.001), which is probably due to the inhibition of efflux transporters rather than metabolism.  Diltiazem did not have any significant effects. Rivaroxaban is a substrate of P-glygoprotein and verapamil and diltiazem are both known inhibitors of P-glygoprotein.  Therefore, concomitant use of rivaroxaban with these non-DHPs might lead to an increase in systemic rivaroxaban exposure and anticoagulant effects.      There is an appropriate Introduction to the study and the Materials and Methods are comprehensive.  The Results are clearly presented and examined and the conclusions made are supported by the results. Recommendation The publishability of this manuscript in Pharmaceutics will be improved following some minor revisions, including those given below. Line 21. Remove “(AUC)” Lines 23/24. Change “in a proportional manner; however, diltiazem did not show any significant effects.” To “in a proportional manner; diltiazem did not show any significant effects, however. Line 35. Change “Xa; it also” to “Xa, it also” Line 39. Change “and treat of deep vein thrombosis” to “and treat deep vein thrombosis” Line 107. Remove “as a reagent” Line 114. Change “by trapezoidal rule” to “by the trapezoidal rule” Line 117. Change “The time courses of prothrombin time” to “The courses of prothrombin time” Line 119. Change “by trapezoidal rule” to “by the trapezoidal rule” Line 140. Change “the time profiles of prothrombin time” to “the profiles of prothrombin time” Lines 203-205. Please explain this better: “Although a multiple-peaks phenomenon was observed in the mean time courses of rivaroxaban plasma concentrations (Figure 3), double peaks are more profound in individual data set and the multiple peaks are a result of inter-individual variability on the times of those peaks.” Line 223. Change “Mean maximum percentage increase” to “The mean maximum percentage increase” Lines 228-229. Change “time curve of prothrombin time” to “curve of prothrombin time” Line 240. Change “straight line” to “solid line” Line 278/279. Change “Because, as shown in Figure 2, the changes of prothrombin time nearly overlapped with the change of corresponding plasma concentrations,” to “Because the changes of prothrombin time nearly overlapped with the change of corresponding plasma concentrations (Figure 2),” Line 283. Change “prothrombin time in healthy pediatric population” to “prothrombin time in a healthy pediatric population” Lines 284/285.  Please explain in more detail how the rescaling was done: “In the present study, prothrombin times are relatively rescaled to allow to direct comparisons with the data obtained by Kubitza et al.” Line 289. Change “intervals” to “interval” Line 307. Change “straight and dashed lines” to “solid and dashed lines” Line 310. Change “fewstudies” to “few studies” The conclusions should state what further studies are required for examining drug-drug interactions between rivaroxaban and verapamil/diltiazem, including examination of the possible roles of efflux transporters in the observed effects.

Author Response

Recommendation The publishability of this manuscript in Pharmaceutics will be improved following some minor revisions, including those given below.

Line 21. Remove “(AUC)”

‘AUC’ was deleted.

Lines 23/24. Change “in a proportional manner; however, diltiazem did not show any significant effects.” To “in a proportional manner; diltiazem did not show any significant effects, however.

The sentence was rephrased accordingly.

Line 35. Change “Xa; it also” to “Xa, it also”

→ ‘;’ was changed to ‘,’.

Line 39. Change “and treat of deep vein thrombosis” to “and treat deep vein thrombosis”

→ ‘of’ was deleted.

Line 107. Remove “as a reagent”

→ ‘as a reagent’ was deleted.

Line 114. Change “by trapezoidal rule” to “by the trapezoidal rule”

→ ‘the’ was inserted.

Line 117. Change “The time courses of prothrombin time” to “The courses of prothrombin time”

→ That should not be changed to make it clear.

Line 119. Change “by trapezoidal rule” to “by the trapezoidal rule”

→ ‘the’ was inserted.

Line 140. Change “the time profiles of prothrombin time” to “the profiles of prothrombin time”

→ That should not be changed to make it clear.

Lines 203-205. Please explain this better: “Although a multiple-peaks phenomenon was observed in the mean time courses of rivaroxaban plasma concentrations (Figure 3), double peaks are more profound in individual data set and the multiple peaks are a result of inter-individual variability on the times of those peaks.”

The sentence was changed as follows: “Although a multiple-peaks phenomenon was observed in the mean time courses of rivaroxaban plasma concentrations (Figure 3), double peaks are more profound in individual data set as shown in Figure 6 and the multiple peaks are a result of inter-individual variability on the times of those peaks.”

Line 223. Change “Mean maximum percentage increase” to “The mean maximum percentage increase”

→ ‘The’ was inserted.

Lines 228-229. Change “time curve of prothrombin time” to “curve of prothrombin time”

→ That should not be changed to make it clear.

Line 240. Change “straight line” to “solid line”

→ ‘Straight’ was changed to ‘solid’.

Line 278/279. Change “Because, as shown in Figure 2, the changes of prothrombin time nearly overlapped with the change of corresponding plasma concentrations,” to “Because the changes of prothrombin time nearly overlapped with the change of corresponding plasma concentrations (Figure 2),”

The sentence was rephrased accordingly.

Line 283. Change “prothrombin time in healthy pediatric population” to “prothrombin time in a healthy pediatric population”

→ ‘a’ was inserted.

Lines 284/285.  Please explain in more detail how the rescaling was done: “In the present study, prothrombin times are relatively rescaled to allow to direct comparisons with the data obtained by Kubitza et al.”

→ The percentage increases of prothrombin time were switched to the ratio of the baseline level, which was added at the end of the sentence: “In the present study, prothrombin times are relatively rescaled to allow to direct comparisons with the data obtained by Kubitza et al., namely, the percentage increases of prothrombin time were switched to the ratio of the baseline level.”

Line 289. Change “intervals” to “interval”

→ ‘s’ was deleted.

Line 307. Change “straight and dashed lines” to “solid and dashed lines”

→ ‘straight’ was changed to ‘solid’.

Line 310. Change “fewstudies” to “few studies”

→ It was changed properly.

The conclusions should state what further studies are required for examining drug-drug interactions between rivaroxaban and verapamil/diltiazem, including examination of the possible roles of efflux transporters in the observed effects.

→ The last sentence of ‘Conclusions’ was revised: “Clinical significance of the present results should be taken into further account including the drug-drug interactions between rivaroxaban and P-gp inhibitors in terms of drug transportations.

Reviewer 4 Report

In this manuscript, the authors investigated the DDI effect of verapamil and diltiazem on the PKPD of rivaroxaban.  The experiments were done on rats. Data were described using a PKPD model.  Significant impact of verapamil on the exposure of rivaroxaban was observed and could be captured by the model.  The PKPD basic structure was sufficient and mechanistically correct. However, several modifications need to be made before considering publishing the work.

In the introduction section, please add some background information on verapamil and diltirazem on their interaction with enzymes and transporters.

Please justify why the number of 15 rats with 5 in each group was used.  

In the experiment design, all the drugs were given as single dose? Please justify whether the reason of not using multiple dose, and whether steady state effect has been reached.

Figure 6 is redundant.  Please either replace that with real population data, or delete it.  Overall time-course was already shown in Figure 7.

A major change should be made to the modeling part: instead of analyzing data by each treatment group separately, combine the data from two groups. DDI effect could be analyzed as a covariate on parameters of interests, such as Ka, F and any other ones that might be mechanistically plausible.  The current discrepancy observed on estimates for apparent volume of distribution might be introduced by the difference in bioavailability between two groups. 

Figure 7 showing that the variability was over estimated, 90% CI should cover 90% of the observation, instead of all of them. This over-prediction might be reduced one you combine all the data, so that a better prediction could be made on inter-individual variability.

Author Response

In this manuscript, the authors investigated the DDI effect of verapamil and diltiazem on the PKPD of rivaroxaban.  The experiments were done on rats. Data were described using a PKPD model.  Significant impact of verapamil on the exposure of rivaroxaban was observed and could be captured by the model.  The PKPD basic structure was sufficient and mechanistically correct. However, several modifications need to be made before considering publishing the work.

In the introduction section, please add some background information on verapamil and diltirazem on their interaction with enzymes and transporters.

→ The following sentence was added in ‘Introduction’, and the literatures are cited accordingly. Non-dihydropyridine calcium channel blockers (non-DHPs), verapamil and diltiazem, are also well-known to be CYP3A4/P-gp inhibitors [6, 7] and have been frequently prescribed in patients with arterial fibrillation [4, 5].”

6. S.F., Zhou. Drugs behave as substrates, inhibitors and inducers of human cytochrome P450 3A4. Curr Drug Metab 2008, 9, 310-322

7. C.E., Frost, W., Byon, Y., Song, J., Wang, A.E., Schuster, R.A., Boyd, D., Zhang, Z., Yu, C., Dias, A., Shenker, F., LaCreta. Effect of ketoconazole and diltiazem on the pharmacokinetics of apixaban, an oral direct factor Xa inhibitor. Br J Clin Pharmacol 2015, 79, 838-846.

Please justify why the number of 15 rats with 5 in each group was used.  

→ In general, five to six animals have been widely used for each group, although the number have to be decided depending on both the inter-individual variability and ethical issues. The numbers of animals can be sufficient enough to present the purposes of this study. 

In the experiment design, all the drugs were given as single dose? Please justify whether the reason of not using multiple dose, and whether steady state effect has been reached.

→ A single dose experiment is good enough to show the inhibitory effects of drugs on enzymes and/or transporters. In addition, as long as the dose is within a range representing a linear kinetics, similar results can be expected even at steady state.

Figure 6 is redundant.  Please either replace that with real population data, or delete it.  Overall time-course was already shown in Figure 7.

→ Figure 6 represents how well the model characterizes the observed individual data, and such figures are frequently shown. In addition, Figure 7 is confidence interval of population predictions by Monte Carlo simulation, not the results identified by the model.

A major change should be made to the modeling part: instead of analyzing data by each treatment group separately, combine the data from two groups. DDI effect could be analyzed as a covariate on parameters of interests, such as Ka, F and any other ones that might be mechanistically plausible.  The current discrepancy observed on estimates for apparent volume of distribution might be introduced by the difference in bioavailability between two groups. 

→ It’s our understanding that a couple of modeling tools, such as a standard two-stage method, a population approach, and Bayesian estimation, may be used to analyze the PK/PD data sets depending on their characteristics. Although one could use a population modeling approach, a standard two-stage method would also be useful for the present data densely observed in a homogenous population. The limitation in terms of over-simplification by a 1-compartment model seems not to be overcome either.

Figure 7 showing that the variability was over estimated, 90% CI should cover 90% of the observation, instead of all of them. This over-prediction might be reduced one you combine all the data, so that a better prediction could be made on inter-individual variability.

→ As mentioned for the above point, Figure 7 is confidence interval of population predictions by Monte Carlo simulation, not the results identified by the model.

Round 2

Reviewer 1 Report

In general, the authors answered partial questions, although there are no additional experiments were conducted, the additional discussions are ok. 

Author Response

In general, the authors answered partial questions, although there are no additional experiments were conducted, the additional discussions are ok. 

→ We believe that no further response would be needed.

Reviewer 4 Report

Figure 6 is redundant.  Please either replace that with real population data, or delete it.  Overall time-course was already shown in Figure 7.

→ Figure 6 represents how well the model characterizes the observed individual data, and such figures are frequently shown. In addition, Figure 7 is confidence interval of population predictions by Monte Carlo simulation, not the results identified by the model.

Goodness-of-fit can not be fully accessed with the fitting curve to a single subject.  Fitting to the overall population and each individual should be provided.

fig 7 is the simulation based on the final model you developed, isn't it?  No matter which way you generate the simulation, after all it reflects how well the model is able to capture the data.

A major change should be made to the modeling part: instead of analyzing data by each treatment group separately, combine the data from two groups. DDI effect could be analyzed as a covariate on parameters of interests, such as Ka, F and any other ones that might be mechanistically plausible.  The current discrepancy observed on estimates for apparent volume of distribution might be introduced by the difference in bioavailability between two groups. 

→ It’s our understanding that a couple of modeling tools, such as a standard two-stage method, a population approach, and Bayesian estimation, may be used to analyze the PK/PD data sets depending on their characteristics. Although one could use a population modeling approach, a standard two-stage method would also be useful for the present data densely observed in a homogenous population. The limitation in terms of over-simplification by a 1-compartment model seems not to be overcome either.

If you are not able to use a population approach, it should be explained why Vd/Fs were different in the two groups, which was not expected. A absolute BA study will help to provide the true F. Then whether Vd differs between the two groups will be clear.

Your last sentence about 1-comp model is confusing.

Figure 7 showing that the variability was over estimated, 90% CI should cover 90% of the observation, instead of all of them. This over-prediction might be reduced one you combine all the data, so that a better prediction could be made on inter-individual variability.

→ As mentioned for the above point, Figure 7 is confidence interval of population predictions by Monte Carlo simulation, not the results identified by the model.

As I said above, no matter what method you use for VPC plots, the prediction interval is an assessment of how well the variability was captured in the model.  Please explain where the huge uncertainty, reflected in the current wide CI, is from.

Author Response

Figure 6 is redundant.  Please either replace that with real population data, or delete it.  Overall time-course was already shown in Figure 7.

→ Figure 6 represents how well the model characterizes the observed individual data, and such figures are frequently shown. In addition, Figure 7 is confidence interval of population predictions by Monte Carlo simulation, not the results identified by the model.

Goodness-of-fit can not be fully accessed with the fitting curve to a single subject. Fitting to the overall population and each individual should be provided. fig 7 is the simulation based on the final model you developed, isn't it?  No matter which way you generate the simulation, after all it reflects how well the model is able to capture the data.

Figure 6 and sentences related are deleted.

A major change should be made to the modeling part: instead of analyzing data by each treatment group separately, combine the data from two groups. DDI effect could be analyzed as a covariate on parameters of interests, such as Ka, F and any other ones that might be mechanistically plausible.  The current discrepancy observed on estimates for apparent volume of distribution might be introduced by the difference in bioavailability between two groups. 

→ It’s our understanding that a couple of modeling tools, such as a standard two-stage method, a population approach, and Bayesian estimation, may be used to analyze the PK/PD data sets depending on their characteristics. Although one could use a population modeling approach, a standard two-stage method would also be useful for the present data densely observed in a homogenous population. The limitation in terms of over-simplification by a 1-compartment model seems not to be overcome either.

If you are not able to use a population approach, it should be explained why Vd/Fs were different in the two groups, which was not expected. A absolute BA study will help to provide the true F. Then whether Vd differs between the two groups will be clear. Your last sentence about 1-comp model is confusing.

We mean that there are three parameters (ka, V/F, ke) which can be dealt with in 1-comp model. As shown in Table 3 and discussed, because there was no change in ke of rivaroxaban when verapamil combined, the differences of rivaroxaban exposure could be only explained by the changes of the other two parameters.  Although verapamil is well known as a strong P-gp inhibitor, the current model could not fully characterize the influence of the drug, which is a limitation of a compartment modeling approach due to the oversimplification of complex systems in vivo.

We fully agree to your opinion: considering a 2.8-fold increase of relative bioavailability, one could get similar volume of distribution, and the 2.6-fold decrease of volume of distribution seems to be inversely reflected by the relative BA.

Figure 7 showing that the variability was over estimated, 90% CI should cover 90% of the observation, instead of all of them. This over-prediction might be reduced one you combine all the data, so that a better prediction could be made on inter-individual variability.

→ As mentioned for the above point, Figure 7 is confidence interval of population predictions by Monte Carlo simulation, not the results identified by the model.

As I said above, no matter what method you use for VPC plots, the prediction interval is an assessment of how well the variability was captured in the model.  Please explain where the huge uncertainty, reflected in the current wide CI, is from.

More or less 30% of relative standard deviations might not be too high. The results by Monte Carlo simulation was obtained by a 1000-repeated simulation with a set of parameters which randomly selected within variance of each parameter. The 90% CI was obtained from the stochastic simulation, which means that the interval does not cover 90% of the data observed.      

Round 3

Reviewer 4 Report

NA